# Entomological Investigation of the Main Entomatic Adversities for Terrestrial Gastropods *Helix aspersa* Müller (Mollusca Gastropoda Pulmonata): A Preliminary Study in Sardinian Heliciculture Farms

**DOI:** 10.3390/insects13070660

**Published:** 2022-07-21

**Authors:** Giulia Murgia, Federica Loi, Stefano Cappai, Maria Paola Cogoni

**Affiliations:** 1Istituto Zooprofilattico Sperimentale della Sardegna, 09125 Cagliari, Italy; giulia.murgia@izs-sardegna.it (G.M.); paola.cogoni@izs-sardegna.it (M.P.C.); 2Osservatorio Epidemiologico Regionale della Sardegna, Istituto Zooprofilattico Sperimentale della Sardegna, 09125 Cagliari, Italy; stefano.cappai@izs-sardegna.it

**Keywords:** agroecology, regulating ecosystem services, insect conservation, snail farming, snail predatory beetle, *Helix aspersa* Müller

## Abstract

**Simple Summary:**

Beetles that predate on the *Helix aspersa* Müller species represent a serious problem for snail farms, causing serious damage, and progressively lead to a reduction in the snail population. Entomological control in snail farms is of fundamental importance to avoid massive colonization that could lead to their death. The greatest difficulty in finding effective solutions to combat predators is that of not being able to use chemicals, given that they would damage the health of the snails. In this study, all the variables that may be considered to avoid an increase in predators were considered and are discussed.

**Abstract:**

In the years 2020–2022, a survey was carried out with the aim of controlling the entomofauna present in seven Sardinian snail farms. The sampling was carried out during the spring–summer and autumn–winter periods, corresponding to the production cycles of the *Helix aspersa* snails (Müller, 1774), the species most frequently bred in snail farms. The samples were taken from farms located in different areas of the region. For the predatory species found in most of the farms, 38% of the individuals were of the *Silpha tristis* Illiger, 1798 (Coleoptera: Silphidae) species, 32% were *Ocypus olens* Müller, 1764 (Coleoptera: Staphylinidae), and 24% were *Carabus* (*Macrothorax*) *morbillosus constantinus* Kraatz, 1899 (Coleoptera: Carabidae). The *Lampyris sardiniae* Geisthardt, 1987 (Coleoptera: Lampyridae) and *Licinus punctatulus* Fabricius, 1792 (Coleoptera: Carabidae) species were equally detected in 3% of the farms. In some farms, the predatory species *S. tristis* and *C.*
*morbillosus costantinus* had caused the death of several snails. This preliminary study aimed to provide a first evaluation and description of the critical issues facing the snails present in seven Sardinian snail farms. First, a specific survey of the entomofauna during two observational periods (i.e., the spring–summer and autumn–winter periods) was conducted. Context-specific description and evaluation will allow providing valid information for Italian and European heliciculture farms to ensure the well-being of the mollusks. The presence of predatory species in snail farms is not easy to control, but some precautions could be useful to avoid massive colonization.

## 1. Introduction

The *Helix aspersa* snail is a terrestrial gastropod mollusk belonging to the Helicidae family, widespread in the Mediterranean basin and in northwestern Europe. It is present throughout Italy and is the most bred species in heliciculture [1].

Heliciculture is a branch of animal husbandry that has, as its objective, the production of snails for food purposes and, secondarily, for the extraction of mucous secretions suitable for the production of products for the parapharmaceutical, medical device, and cosmetic sectors. The breeding systems in heliciculture are represented by outdoor plants (outdoor) or indoor plants (indoor); in Italy, they are mainly outdoor plants on open land [1,2].

The *H. aspersa* species, also known as *Cornu aspersus*, possesses a marked adaptability in captive conditions, such as breeding helicopters; therefore, it is considered the most suitable species for helicopter activity [3]. The snail is an insufficient hermaphrodite, and thus it needs to mate with another subject, as it cannot self-fertilize. After a courting phase, the snail mates by extending the pouch containing the dart, which solidifies upon contact with the air, allowing mutual solicitation; the genital orifice is pushed outwards, and mating takes place. The *H. aspersa* species lays about 60–90 spawning eggs in a hole dug in the ground. The deposition generally takes place twice a year, but sometimes, if the conditions are favorable, it can occur three or four times a year. Embryogenesis lasts about 15–25 days, and the small snails that emerge are identical to the adult but small in size (about 3 cm). During the early stages of growth, their weight increases significantly until they reach genital development, at which point growth slows [1,2].

In nature, snails can encounter various difficulties due to food shortages, periods of drought, or unfavorable temperatures; to survive, they need rest periods, as they need to recover from both these unfavorable factors and from mating periods. On the farm, it is essential to observe these rest periods to preserve their survival beyond 14 months [1]. The periods of quiescence (hibernation during the winter period and aestivation during the summer period) are facilitated by producing a light veil around the opening of the shell, which stops evaporation. If the period of inactivity is long, they produce a real calcareous operculum that helps the snail to defend itself from climatic adversities and predators. These species generally pass periods of low temperature in shelters attached to walls, stones, or wood. Once conditions favorable for the life of the snail have been restored, the awakening phase takes over, in which the snail resumes feeding before the mating phase [1,2].

The main critical issues for snail farms are environmental conditions that could affect both plant production and the well-being of the snails. The habitat in which they live is of fundamental importance for their survival and, consequently, for the success of the breeding. Among the main critical issues are the biotic adversities, which include the predatory, phytophagous, and parasitic entomofauna that alternate within the snail farms and are attracted to different sources of nutrition. The arthropod families that mainly feed on terrestrial gastropods are Staphylinidae, Lampyridae, Carabidae, Silphidae, and Drilidae [4]. 

Carabid Beetles and Staphylinidae have been identified as the most frequent predators of terrestrial gastropods [4,5,6]. Particularly, Staphylinides are considered the main enemies of snails [1,7,8,9]. The larval stage of the Lampyridae family is considered a snail predator [4,10,11]. Some species belonging to the Silphidae family are both adult and larval snail predators [4]. Most predation-related factors are associated with the population density, intra- and inter-specific predators, the availability of live and dead gastropods, and the presence of food wastes [4].

The knowledge on the predatory arthropod fauna present in snail farms remains insufficient, and most of the studies have been carried out on specimens in the wild [12,13]. 

Some of these studies carried out on other species of snails have analyzed the lesions present on the shells in an attempt to trace the predator [12]; to date, the extent to which predatory actions affect the snail population and the selection of prey by predators and parasites have not been confirmed [13]. Furthermore, there is a lack of specific data on the predatory species found in the Italian territory. Most of the studies focus on the snail’s role as a harmful parasite for agriculture. Snails constitute a serious menace to agricultural production, resulting in significant economic losses for a wide range of crops (e.g., rape, vegetables, legumes, cereals, and fruits) [14]. The welfare and health of bred gastropod mollusks (snails and slugs) have not been studied in depth.

This study aimed to provide a preliminary assessment and description of the critical issues facing the snails present in seven Sardinian snail farms. First, a specific survey of the entomofauna during two observational periods (i.e., the spring–summer and autumn–winter periods) was conducted. The context-specific description and evaluation will allow providing valid information for Italian and European heliciculture farms to ensure the well-being of their mollusks.

## 2. Materials and Methods

This study on predatory entomofauna was carried out in seven snail farms included in the HelixREC project (https://www.izs-sardegna.it/helixrec/, accessed on 20 July 2020). It was conducted during the two typical reproductive cycles of the snails (spring–summer and autumn–winter), with the main aim of collecting evidence on the predatory and phytophagous entomological species present in Sardinian helicopters.

The farms were located in different southern areas of Sardinia (Figure 1). All the farms were in free lands defined as “outdoor” [1] and used the full biological cycle breeding technique. Fourteen samples were carried out, two per farm, during the two different seasons (i.e., spring–summer and autumn–winter). Farms were characterized by an average surface of 1645 m^2^ (SD = 572) and about 25% of each area was sampled (mean sampled surface = 411, SD = 143). Some farm areas were divided into different rows delimited by anti-escape nets, while others also had external perimeter fences as a further barrier to contain the snails and to avoid incursions by predators. There were also farms equipped with superior nets to protect the farm from hail and birds. The farms’ characteristics were collected in detail by farm code. The seven farms were evaluated based on the sampled surface (m^2^), the type of fence (i.e., a solid barrier, electric fence, or simple metal fence), the presence of rows singularly fenced or not, and the presence of superior nets.

### 2.1. Sampling Period

The sampling periods used in the helicopter farms were conditioned by several factors. In Sardinia, the temperatures become very high during the summer season, particularly in July–August. The sudden change in temperature forces the snails into a longer and earlier hibernation, reducing the time for sampling. The biological cycle of the snails, as mentioned, is temperature-dependent and, therefore, conditioned by the external temperature.

At the end of the biological cycle of the snails both during the spring/summer season and during the autumn/winter season, the specimens present in the rows are taken: some are put up for sale, while the others that have good physical characteristics are inserted into the rows as reproducers for the next production cycle.

In each of the farms, the sampling was performed during the main slag reproductive periods (i.e., spring–summer and autumn–winter) between 2021 and 2022. For the spring–summer sampling, the reference months were May, June, and the first week of July, while the autumn–winter sampling was carried out in October, November, and December. The entomological investigations were all conducted during the morning, between 9:00 am and 13:00 pm, to survey the predatory, phytophagous, and parasite entomofauna present. Visual observation for at least 4 h was carried out in the whole farm’s area along with the sampling.

### 2.2. Sampling Techniques

All the captures were carried out using different entomological equipment (i.e., entomological nets with mowing and moths, and entomological tweezers for manual sampling). Tests were conducted using pitfall traps equipped with superior nets with meshes of different sizes in some farms. However, this method was ruled out because it was found that numerous young snails died inside the traps. In addition, during irrigation in the rows, the traps constantly filled with water, causing the trigger liquid to escape despite the protectors positioned above the traps. Other types of capture were excluded, as any system would have disrupted the daily management activities on the farm. In addition to the reasons set out, it is necessary to consider the primary importance of the daily water supply, which does not allow for solutions for the positioning of the traps.

For each sampling, a standard area corresponding to at least 25% of the farm’s active area was assessed, including specific rows reserved for the project. During the on-field activities, all the samples taken were placed in containers provided with chopped cork soaked in ethyl acetate to avoid losses due to predation during transport. The samples were subsequently transferred to tubes containing 70% ethanol. 

The taxonomic determination was performed by macro-microscopic morphological examination with the aid of a stereomicroscope and entomological dichotomous keys [15,16,17,18,19]. Considering that the presence of phytophagous species did not create problems for the snails in the involved farms, in this study, we only considered the predatory species.

### 2.3. Statistical Analyses

The completeness and consistency of the collected data, which were stored in an ad hoc database, were evaluated. Descriptive statistical analyses were carried out. The quantitative variables were summarized as mean values, standard deviations (SDs), medians, and interquartile ranges (IQRs), whereas the qualitative variables were summarized as frequencies and percentages. To compare the qualitative variables, either the Chi-square test or the Fisher’s exact test was applied. The Kruskal–Wallis nonparametric test was applied to compare differences between quantitative variables. Furthermore, to identify the characteristic of the farms more related to the increasing number of insects detected, the negative binomial regression model was applied using the number of insects detected as the outcome. Negative binomial regression was chosen rather than Poisson regression based on the low number of included farms, the overdispersion of the data, and the lower values of the Bayesian Information Criterion (BIC) and the Akaike Information Criterion (AIC) generated by this model. Candidate explicative variables were selected if they were significant in the univariate test or for their clinical relevance. Results were reported as adjusted Odds Ratios (ORs) with corresponding 95% Confidence Intervals (95% CI). Goodness of fit was assessed based on the R^2^ adjusted value. The level of *p* < 0.05 was considered significant for all the analyses. The software employed to carry out the analysis was STATA/SE for Windows, version 15 (Stata Corp., College Station, TX, USA).

## 3. Results

A total of seven snail farms mainly located in the southeast of the Sardinian region were included in this study. Three farms presented solid barriers in the perimeter (i.e., metal fence) and only one farm was characterized by an electric fence (Sf2). In most of the farms (5, 71%), the rows were divided by a single fence and presented a superior net. The main features related to each observed farm are reported in Table 1. 

Sf1 farm exhibited total coverage and presented division into internal rows. The perimeter of this farm was delimited by plexiglass panels and polyethylene nets with tight meshes on all sides and on the top. Internally, it was divided into rows or internal enclosures delimited, in turn, by anti-escape nets with double flounces in polyethylene; between one row and the other, one could observe the bare dirt walkway area (Figure 2a,b).

Sf2 farm was characterized by a single fence with a double-flounce electrified perimeter net and a metal fence (Figure 2c). This farm and the Sf6 farm were the only ones that did not have full farm coverage.

In Sf3 farm, there was no perimeter fence but there was a subdivision into rows with total coverage of the same. Narrow-mesh polyethylene nets were embedded in both the right and left bases. There was no area intended for walking.

Sf4 farm, as well as Sf1, had total coverage and internal subdivision of the rows (see Sf1), and the perimeter was delimited by galvanized ondulin. The rows were further delimited by narrow-mesh polyethylene nets, divided by walkways on bare ground.

Sf5 was characterized by total coverage with high polyethylene nets with narrow meshes; internally, the rows were not delimited by further nets but appeared as a single area, separated from walkways with mulch sheets.

Sf6 had a perimeter delimitation in galvanized ondulin and internal subdivision into rows further delimited by polyethylene nets with tight meshes and double flounces, and the walkways were bare in beaten earth. Sf3 and Sf7 were the only two farms characterized by single rows completely covered by net (Figure 2d).

Table 2 summarizes the data for the entomological samples of the predator species collected during the two reproductive periods (Time 1, spring–summer 2021; Time 2, autumn–winter 2021; Time 3 spring–summer 2022).

Farms Sf3 and Sf4 were the most populated by predators of different species. In Sf3, more than 200 predators were found, mainly during time 1 and time 3 (spring–summer seasons), and more than 97 predators were detected in Sf4 during time 1.

The most frequent species of predators detected in the Sardinian heliciculture farms was the *Silpha tristis* Illiger, 1798 (Coleoptera: Silphidae). More than 50 predators of this species were detected in farms Sf3 and Sf4 during time 1, 20 in Sf3 during time 3, and a small number in Sf2 and Sf5 (*n* = 5 and 2, respectively). No one sample was detected in Sf1, Sf6, and Sf7. A total of 117 predators of the species *Ocypus olens* Müller, 1764 (Coloptera: Staphylinidae) were detected in all the farms except for Sf5, with a maximum number in sf4 during time 1. *Carabus* (*Macrothorax*) *morbillosus costantinus* Kraatz, 1899 (Coleoptera: Carabidae) abundance detection was similar to *Ocypus olens* Müller, 1764 (Coloptera: Staphylinidae), with more than 114 predators, of which at least 50 were collected in Sf3 during time 3. The numbers of the species *Lampyris sardiniae* Geisthardt, 1987 (Coleoptera: Lampyridae) and *Licinus punctatulus* Fabricius, 1792 (Coleoptera: Carabidae) detected were too low to be relevant. 

Furthermore, considering losses associated to predators, in the Sf1 farm, despite the low number of predators sampled (*n* = 7), the presence of the species *C. morbillosus* was confirmed by the massive death of the snails observed in time 3 and the numerous shells found during the sampling that presented the characteristic defined as “shell crushing”. In fact, the individuals belonging to the Carabidae family typically caused damage in terms of shell crushing, starting from the opening of the shell and continuing along the line of the intrasutral zone defined as “body whorl” [5]. In Sf3, the overall presence of the entomofauna was significant but never caused significant damage to the snail population. The massive infestation of *Silpha tristis* (*n* > 50) in the Sf4 farm caused the death of most of the snails. In Sf5 and Sf7, the numerical predator counts were negligible (*n* = 16, *n* = 2, respectively). In Sf2 and Sf6, the only two farms without superior nets, damage from predators (i.e., entomofauna or vertebrates) was never detected.

Most of the predators were detected during the spring–summer period (*n* > 346, 89%) rather than autumn–winter (*n* = 42, 11%). To avoid sampling bias, considering the different proportion of sampling (2:1) and the critical climate conditions associated to May–July 2022 with not normal average temperatures, which probably caused the death of all the snails in the two farms, only time 1 and time 2 sampling periods were considered for statistical analysis. A statistically significant higher number of insects was detected in those farms characterized by the presence of a perimeter metal fence (*p* = 0.021) and a borderline difference was detected in those with a superior net (*p* = 0.48). This difference was particularly evident in the number of *Lampyris sardiniae* Geisthardt, 1987 (Coleoptera: Lampyridae) and *Licinus punctatulus* Fabricius, 1792 (Coleoptera: Carabidae).

The results of the regression model reported in Table 3 highlight that the total surface area slightly increased the detection of insects: every 100 m^2^ of surface, the number of insects detected increased by 12.5% (OR_adj_ = 1.125 (95% CI = 1.237–1.586), *p* = 0.043). 

The presence of a solid barrier around the farms increased the number of predators detected five times (OR_adj_ = 5.251 (95% CI = 1.865–27.569), *p* = 0.001). The number of insects increased about three times if the farm was characterized by a superior net (OR_adj_ = 2.615 (95% CI = 1.369–4.994), *p* = 0.011). Furthermore, the probability to detect an increasing number of predators is strictly affected by the season: particularly, during the spring–summer period, the number of insects was about five times higher with respect to autumn–winter (OR_adj_ = 4.498 (95% CI = 1.269–15.943), *p* = 0.011). Otherwise, considering the very low simple size, the goodness of fit of the negative binomial regression model was medium–low (R^2^ adjusted = 0.43).

## 4. Discussion

The presence of predatory insects on farms is a significant problem for farmers. There are situations of equilibrium in which the predatory entomofauna does not cause such damage as to require intervention, but taking measures is necessary in other situations. All the farms included in this project follow a complete natural cycle farming system, and it is essential that the solutions to the problem of the predatory entomofauna respect the environment and snails. 

All the predatory insects were sampled both at the larval stage and at the adult stage. Only the *Lampyris sardiniae* species was sampled exclusively during the larval stage, given that it has helicophageal habits only at the larval stage [4,11]. The presence of both larvae and adults of these species demonstrated that the heliciculture farms are a favorable environment in which to reproduce.

The size of the snails according to the stage of development did not influence the prognosis, as the adult and larval stages of the different beetle species were observed preying on snails of different ages and sizes. To the best of our knowledge, current scientific opinion does not provide clarification on this matter. 

All the included farms greatly suffered from the presence of predators. Two farms (Sf3 and Sf4), both characterized by the massive presence of the beetle, were affected by the presence of *S. tristis*. This species is generally described as a predator with necrophagous habits [20], but in some of the helicopters where the entomological investigation was carried out, it was observed that it had strong predatory tendencies towards snails. In particular, an important mass-death of snails was found in the Sf4 farm, presumably attributable to the *S. tristis* species; to exclude other causes of death, parasitological and microbiological analyses were performed in meat and irrigation water, all yielding negative or negligible results. In the Sf3 company, on the other hand, the massive presence of the same species was promptly stemmed by manual observation and elimination, reducing the predatory action of this species and, consequently, preventing the death of snails. 

On the other hand, in the Sf4 farm, a specific intervention was carried out by installing gravity traps triggered with wine vinegar and water outside the rows. This measure initially obtained excellent results, but the management of the traps presents considerable difficulties due to the frequent emptying and topping up operations. Furthermore, the positioning of the traps within the rows presents the same problem previously described for sampling. These reasons led the farm to abandon this method; unfortunately, manual elimination was not possible due to the large size of the farm. It is clear that if containment measures to combat predators are not implemented, the economic damage can be significant. 

It should also be taken into consideration that both farms have full farm coverage. As demonstrated by the statistical analyses, the number of predators increases by more than five times if solid barriers around the farms are present and about three times more if the farm was characterized by a superior net. Sf3 farm was characterized by a perimeter fence with an upper anti-hail net and, internally, a division into rows, while the farm Sf4 has single rows equipped with lateral and upper nets. Among the sampled farms, those without full coverage (Sf2 and Sf6) have never recorded problems related to entomofauna. The nets present in the farms are used as a barrier for the containment of snails, as well as for protection against meteorological and biotic adversities. From the entomological point of view, nets do not always guarantee an impenetrable barrier for the predators, but it is possible that, once insects go inside, they will no longer be able to get out. Furthermore, for predatory insects, snail farms represent an ideal habitat given the high availability of snails, phytophagous insects, and other arthropods. 

Another element to consider is the superior nets (i.e., anti-hail and anti-bird) that prevent the entry of vertebrates such as birds considered enemies of snails. However, most birds are predators of beetles such as Carabidae, Staphylinidae, and Silphidae. Recently, a study on the role of beetles in the diet of raptors highlighted that the insects belonging to this order were the main category of prey (66.5% for the Common Kestrel *Falco tinnunculus*, 67.2% for the Tawny Owl *Strix aluco* (not present in Sardinia), 70.2% for the Little Owl *Athene noctua,* and 63.6% in the Barn Owl Tyto alba) [21]. This could represent another reason why there are few predatory insects in snail farms without upper nets. In general, to further avoid an increase in the predatory arthropod fauna inside the rows, it would be advisable to eliminate the non-viable snails, to avoid the call of predatory and opportunistic species attracted by the odors of decomposition (Staphylinidae, Silphidae, and Diptera), considering that, in addition to feeding on dead snails, they are species capable of attacking and killing their live counterparts [15,16,17].

The beetles called helophages are actually polyphagous species that do not feed exclusively on gastropods but also feed on other prey. Among the causes that could affect their predation, a key role could be played by the population density, the intra- and interspecific competition between predators, and the availability of live and dead gastropods [4]. It is necessary to highlight that the heliciculture farms involved in this study presented a high population density: this could be the reason for the frequent detection of predators belonging to the order of beetles. Even if not directly measured, the size of the farms could be used as a proxy of the population density, thus the statistical analyses confirmed this hypothesis. Furthermore, the availability of the source of food causes predators to enter inside the rows to prey almost exclusively on snails. From an entomological point of view, to ensure the containment of predatory insects, it is necessary to keep the areas surrounding the farm and the walkways between the rows free of vegetation, since the areas where spontaneous plants are present could be an ideal habitat for predatory insects such as beetles or other invertebrates and small vertebrates that could cause damage to the farm. These areas where insects can find refuge are called beetle banks, due to the benefits they provide to insects [22,23]. 

Insects defined as helicophages are considered useful in organic farming, as they play a very important role in the natural fight against insects and other phytophagous invertebrates, such as gastropods. All these studies on the role of predatory insects have been carried out on agricultural ecosystems [24,25,26] because terrestrial gastropods are a serious threat to agricultural crops.

Another factor that could increase the presence of predators on the farm is the presence of shelters for snails. These structures are frequently located inside the rows to protect the snails and to facilitate their collection, given that the snails tend to anchor themselves to these supports to find shelter and protection. The creation of artificial shelters has been shown to provide potential for predator survival [27]. In the seven Sardinian farms, these shelters were made up of wooden boxes or cork panels, slightly raised from the ground. To overcome this problem, the authors suggest using vertical structures that do not create an environment favorable to insects. 

## 5. Conclusions

This study made it possible to elucidate the predatory species most present in the Sardinian helicopter farms examined and to understand what preventive measures could be implemented to stem infestations and the consequent economic damage in the farm.

Taking into account what has been mentioned previously, it would be appropriate to pay attention to some actions: observe the cleaning of the surrounding land and not just cleaning between the rows, because they represent an ideal environment for predators, modify the structures used for the shelter and collection of the snails, preferring vertical structures that do not create an ideal niche for predators, and eliminate dead snails from the rows, which could attract necrophagous predatory species, including snails.

The total coverage with the networks shown for some plants was associated with the presence of a greater number of predators. Therefore, it would be necessary to continue with further investigations to better determine the role that these cover systems play regarding predators. It might be interesting to conduct new studies that explore new containment techniques for predatory insects, natural enemies of snails, from the perspective of corporate sustainability, to allow the farmer to intervene even in the presence of snails without causing damage to them.

## Figures and Tables

**Figure 1 insects-13-00660-f001:**
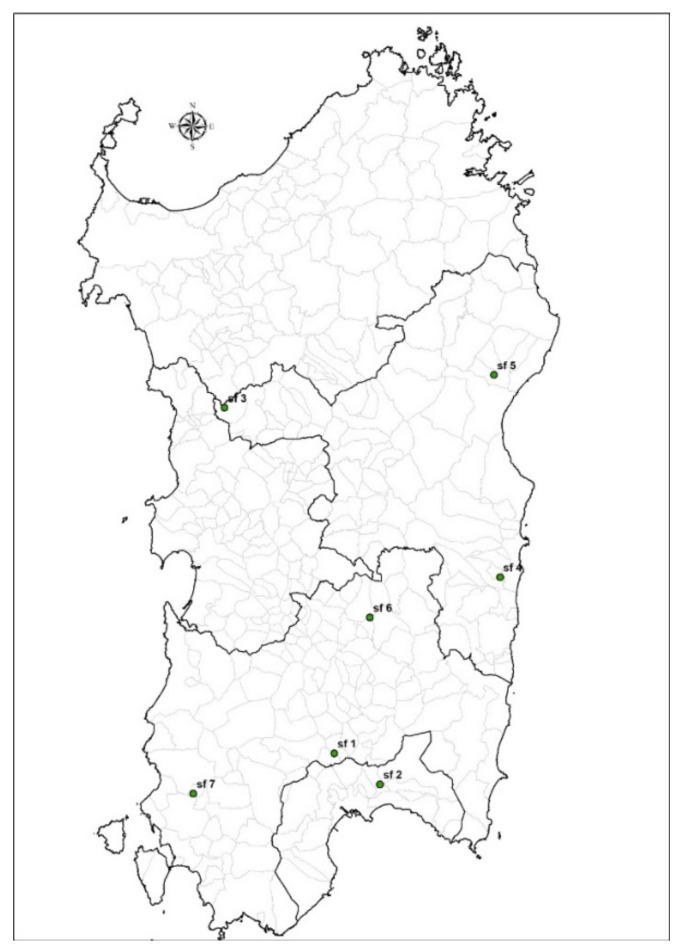
Geographical locations of the seven snail farms.

**Figure 2 insects-13-00660-f002:**
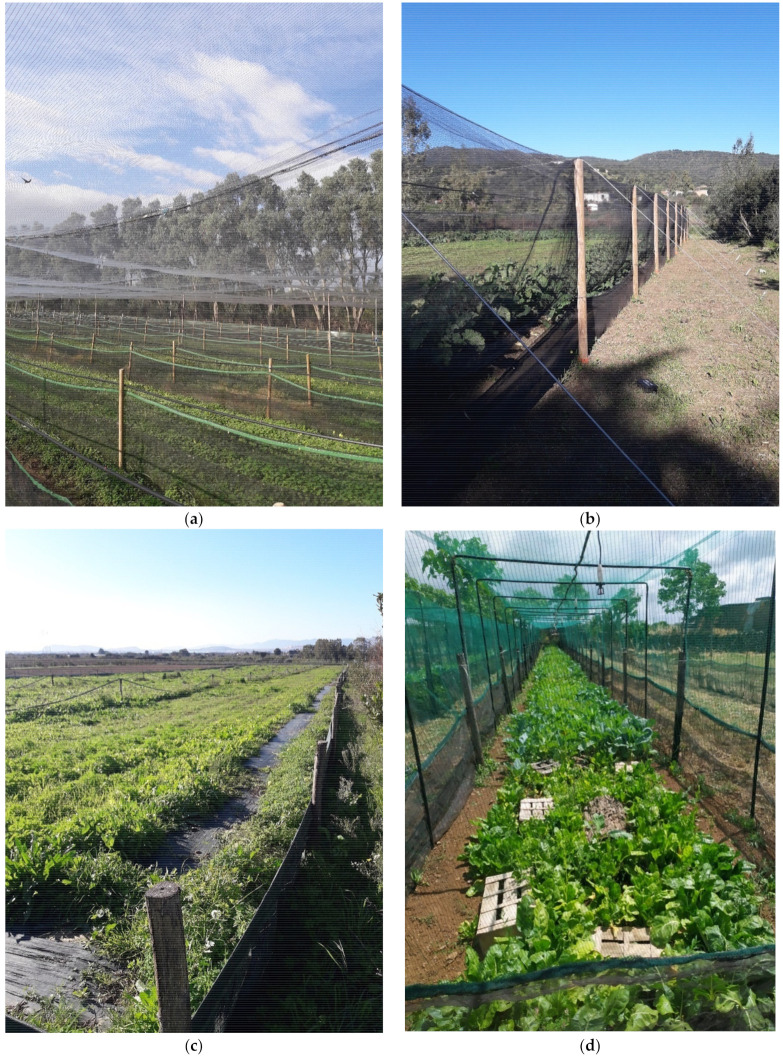
Types of fences and rows in the seven farms involved in this study: (**a**) single-fenced row with unique superior net, (**b**) simple perimeter fence, (**c**) unique rows with perimeter metal fence without net, and (**d**) single-fenced row with single superior net.

**Table 1 insects-13-00660-t001:** Baseline characteristics of the seven involved farms.

Snail Farm Code	Total Surface (m^2^)	Sampled Surface (m^2^)	Type of System	Solid Barrier	Electricity Fence	Single-Fenced Rows	Superior Net
Sf1	1350	338 (25%)	Outdoor	Yes	No	Yes	Yes
Sf2	2200	550 (25%)	Outdoor	No	Yes	No	No
Sf3	1170	293 (25%)	Outdoor	No	No	Yes	Yes
Sf4	2520	630 (25%)	Outdoor	Yes	No	Yes	Yes
Sf5	1350	338 (25%)	Outdoor	No	No	No	Yes
Sf6	1924	481 (25%)	Outdoor	Yes	No	Yes	No
Sf7	1000	250 (25%)	Outdoor	No	No	Yes	Yes

**Table 2 insects-13-00660-t002:** Entomological samples collected during the two time periods in the seven involved Sardinian heliciculture farms, by both manual and optical counting. Data are presented as number (percentage). Time 1 refers to the spring–summer season; time 2 refers to the autumn–winter season.

Snail Farm Code	*Silpha tristis*	*Ocypus olens*	*Carabus morbillosus costantinus*	*Lampyris sardiniae*	*Licinus punctatulus*	Total
Sf1—time 1	0 (0)	0 (0)	2 (1.8)	0 (0)	0 (0)	2 (0.5)
Sf1—time 2	0 (0)	2 (1.7)	3 (2.6)	0 (0)	0 (0)	5 (1.3)
Sf1—time 3	all snails died					
Sf2—time 1	5 (3.9)	4 (3.4)	0 (0)	2 (10.5)	0 (0)	11 (2.8)
Sf2—time 2	0 (0)	0 (0)	0 (0)	0 (0)	0 (0)	0 (0)
Sf2—time 3	0 (0)	3 (2.6)	0 (0)	6 (31.6)	0 (0)	9 (2.3)
Sf3—time 1	>50 (39.4)	23 (19.7)	31 (27.2)	0 (0)	0 (0)	>104 (26.8)
Sf3—time 2	0 (0)	9 (7.07)	0 (0)	0 (0)	1 (9.1)	10 (2.6)
Sf3—time 3	20 (15.7)	30 (25.6)	>50 (43.9)	0 (0)	0 (0)	>100 (25.8)
Sf4—time 1	>50 (39.4)	37 (31.6)	10 (8.8)	0 (0)	0 (0)	>97 (25.0)
Sf4—time 2	0 (0)	4 (3.4)	5 (4.4)	0 (0)	0 (0)	9 (2.3)
Sf4—time 3	all snails were					
Sf5—time 1	0 (0)	0 (0)	4 (3.5)	0 (0)	0 (0)	4 (1.0)
Sf5—time 2	0 (0)	0 (0)	0 (0)	5 (26.3)	7 (63.3)	12 (3.1)
Sf5—time 3	2 (1.6)	0 (0)	0 (0)	6 (31.6)	3 (27.3)	11 (2.8)
Sf6—time 1	0 (0)	4 (3.4)	2 (1.8)	0 (0)	0 (0)	6 (1.5)
Sf6—time 2	0 (0)	0 (0)	6 (5.3)	0 (0)	0 (0)	6 (1.5)
Sf6—time 3	all snails were in aestivation					
Sf7—time 1	0 (0)	1 (0.9)	1 (0.9)	0 (0)	0 (0)	2 (0.5)
Sf7—time 2	0 (0)	0 (0)	0 (0)	0 (0)	0 (0)	0 (0)
Sf7—time 3	0 (0)	0 (0)	0 (0)	0 (0)	0 (0)	0 (0)
Total	>127 (32.7)	117 (30.2)	>114 (29.4)	19 (4.9)	11 (2.8)	>388 (100)

**Table 3 insects-13-00660-t003:** Increasing number of insects detected: adjusted Odds Ratios (OR_adj_) and 95% Confidence Intervals (95% CI), with associated *p*-values from the negative binomial regression model fitted using the total number of insects as the outcome.

Explicative Variables	OR_adj_ (95% CI)	*p*-Value
Total surface (100 m^2^)	1.125 (1.237–1.586)	0.043
Solid barrier or metal fence (yes vs. no)	5.251 (1.865–27.569)	0.001
Superior net (yes vs. no)	2.615 (1.369–4.994)	0.011
Season (spring–summer vs. autumn–winter)	4.498 (1.269–15.943)	0.020

## Data Availability

All data are reported in the main text.

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
