# Peer review of "Entomological Investigation of the Main Entomatic Adversities for Terrestrial Gastropods Helix aspersa Müller (Mollusca Gastropoda Pulmonata): A Preliminary Study in Sardinian Heliciculture Farms"

_insects, 2022, doi:10.3390/insects13070660_

Round 1

Reviewer 1 Report

  1. The abstract is very short and does not contain information about the results obtained. I propose to completely redo the abstract.
  2. The introduction contains incomplete information. The authors gave very little information about the biology of the snail. Also, the authors in the manuscript did not show many publications about snail predators. Here are examples of publications, of which there are a lot in the public domain (doi:10.4039/Ent1071111-10; https://doi.org/10.1016/S0045-6535(02)00116-9; https://doi.org/10.1046/j.1365-2311.2001.00365.x; https://doi.org/10.3157/0013-872X(2006)117[545:PBOGBC]2.0.CO;2).
  3. Line 65-66. If the farms differed in some conditions, then it is necessary to describe all these conditions. This is necessary to understand the possible effects of such conditions on the results obtained.
  4. Line 71. The authors indicated that there is a table in Excel. Where is this table? If the authors meant Table 1 by this table, they should have quoted it in the text and made a description.
  5. Line 75-77. The authors did not set pitfall traps for catching insects. However, it is well known that a lot of ground beetles and staphylinids fall into such traps, some of which are predators. The absence of such traps may give insufficient results. The authors should consider other options for collecting insects, for example, light trapping, the use of barrier traps and other research methods.
  6. The authors did not indicate which manuals they used to identify insects.
  7. The authors did not use any statistical methods, which is unacceptable in environmental studies.
  8. Figure 1 is superfluous and it should not be in the results.
  9. The results contain very little information. From all the results obtained, we can only conclude that there are five species of predators on farms.
  10. The minimum information received (only 260 specimens of predators) does not allow us to draw any conclusions.
  11. In the discussion, the authors describe well-known facts. Nothing new and interesting has been revealed or received.
  12. There is no conclusion.

Author Response

  • The abstract is very short and does not contain information about the results obtained. I propose to completely redo the abstract.
  • R: Dear review, following your suggestion the abstract has been implemented as weel as the whole manuscript. Furthermore, the manuscript has been completely reviewed by the MDPI English editing service.
  • The introduction contains incomplete information. The authors gave very little information about the biology of the snail. Also, the authors in the manuscript did not show many publications about snail predators. Here are examples of publications, of which there are a lot in the public domain (doi:10.4039/Ent1071111-10; https://doi.org/10.1016/S0045-6535(02)00116-9; https://doi.org/10.1046/j.1365-2311.2001.00365.x; https://doi.org/10.3157/0013-872X(2006)117[545:PBOGBC]2.0.CO;2).
  • R: Dear review, following your suggestion the introduction has been implemented including several information on biology of the snail and the references that you indicated.
  • Line 65-66. If the farms differed in some conditions, then it is necessary to describe all these conditions. This is necessary to understand the possible effects of such conditions on the results obtained.
  • R: In lines 129 -152 the authors added several information and description of the farmers
  • Line 71. The authors indicated that there is a table in Excel. Where is this table? If the authors meant Table 1 by this table, they should have quoted it in the text and made a description.
  • R: Dear review, it was a refuse. We have removed the sentence “in a specific Excel spreadsheet” from the text and Table 1 has been quoted
  • Line 75-77. The authors did not set pitfall traps for catching insects. However, it is well known that a lot of ground beetles and staphylinids fall into such traps, some of which are predators. The absence of such traps may give insufficient results. The authors should consider other options for collecting insects, for example, light trapping, the use of barrier traps and other research methods.
  • R: In lines 175-192 (section 2.2 sampling tecniques) the reasons for what no traps were used has been specified “Tests were conducted using pitfall traps equipped with superior nets with meshes of different sizes in some farms. However, this method was ruled out because it was found that numerous young snails died inside the traps. In addition, during irrigation in the rows, the traps filled with water constantly, causing the trigger liquid to escape despite the protectors positioned above the traps. Other types of capture were excluded, as any system would have disrupted the daily management activities on the farm. In addition to the reasons set out, it is necessary to consider the primary importance of the daily water supply, which does not allow for solutions for the positioning of the traps.”
  • The authors did not indicate which manuals they used to identify insects.
  • R: In line 206 the bibliography of reference used to identify insects was specified “The taxonomic determination was performed by macro-microscopic morphological examination with the aid of a stereomicroscope and entomological dichotomous keys [15-20].”
  • The authors did not use any statistical methods, which is unacceptable in environmental studies. Figure 1 is superfluous and it should not be in the results.
  • R: Dear review, considering the very low number of farms included in this study (7), inference analysis is not strongly suggested. Otherwise, following your comments, the statistical analysis paragraph was added in material and methods and in results. Furthermore, statistical analysis was carried out to first investigate the possibile factors associated with the increasing number of insect detected. Figure 1 was moved in material and methods to provide an overview of the farm’s location in the whole region.
  • The results contain very little information. From all the results obtained, we can only conclude that there are five species of predators on farms.
  • R: the manuscript is a first step study to in deep evaluate the predators species in the heliciculture Sardinian farms providing useful information to ensure the welfare of the snails avoiding losses. The results has been strongly implemented.
  • The minimum information received (only 260 specimens of predators) does not allow us to draw any conclusions.
  • R: The results and conclusion have been strongly implemented.

  • In the discussion, the authors describe well-known facts. Nothing new and interesting has been revealed or received.
  • R: The results and discussion have been implemented, otherwise we do not agree with this comment. Helicopter farms are an expanding reality worldwide, given the high consumption of snails by the population. In the literature there are no studies on the measures to be taken to avoid the incursion by entomofauna predating snails. There are studies that analyze the predatory modalities of some species in particular such as Ocypus olens, or other species different from those found in Sardinia. All the other studies were carried out on snails in the wild. Please, if the review had some further suggestion to improve the paper we are ready to follow.
  • There is no conclusion.
  • R: conclusions have been added

Reviewer 2 Report

I would like to thank you for the opportunity to prepare the review for the paper titled “Entomological investigation on the main entomatic adversities in terrestrial gastropods Helix aspersa Müller (Mollusca Gatropoda Pulmonata) in Sardinian heliciculture farms.”

The subject is very interesting and rather new. However, the manuscript would benefit to include more data and statistical analysis (e.g. for the data in table 2).

The paper is a good start but I strongly suggest improving it with more field data. In addition, would be advisable to try to correlate the presence of snail predators with other variables e.g. climate data, environmental parameters, age of the snails, etc…

The manuscript is well done but in a preliminary form. In addition, the authors should report also other data other like economic data and impact on the production of Helix, and how much is lost due to insect predators.

Another really small remark is to add the name of the scientist and the order and family of all the species e.g. Carabus (Macrothorax) morbillosus Kraatz, 1899 (Coleoptera: Carabidae). This should be done for all species of animals

Author Response

  • I would like to thank you for the opportunity to prepare the review for the paper titled “Entomological investigation on the main entomatic adversities in terrestrial gastropods Helix aspersa Müller (Mollusca Gatropoda Pulmonata) in Sardinian heliciculture farms.”
  • The subject is very interesting and rather new. However, the manuscript would benefit to include more data and statistical analysis (e.g. for the data in table 2).
  • R: dear review, thank you very much for the time spent for this revision. Considering the very low number of farms included in this study (7), inference analysis is not strongly suggested. Otherwise, following your comments, the statistical analysis paragraph was added in material and methods and in results. Furthermore, statistical analysis was carried out to first investigate the possibile factors associated with the increasing number of insect detected.
  • The paper is a good start but I strongly suggest improving it with more field data. In addition, would be advisable to try to correlate the presence of snail predators with other variables e.g. climate data, environmental parameters, age of the snails, etc…
  • R: dear review, this is a first step analysis to wich further investigation will follow. Considering your concern, we changed the title of the manuscript including the specification to a prelimary phase of the work. Further variables will be considered and described when several farms will be detected, considering that only 7 farms are not enought (as demonstrated by the statistical analysis). Specification on the age of the snails has been added in lines 320-323
  • The manuscript is well done but in a preliminary form. In addition, the authors should report also other data other like economic data and impact on the production of Helix, and how much is lost due to insect predators.
  • R: as underlined, this is a preliminary study. Some consideration about economic losses has been added in lines 284-303
  • Another really small remark is to add the name of the scientist and the order and family of all the species e.g. Carabus (Macrothorax) morbillosus Kraatz, 1899 (Coleoptera: Carabidae). This should be done for all species of animals
  • R: modifications reported in lines 23-26, as suggested

Reviewer 3 Report

I think this article would be interesting for readers, not so much for snail farmers, but more for those interested in tackling mollusc damage in agrolandscapes in search of effective biological control measures. I suggest a complete change in the idea of the article, from avoiding damage to snails on farms to promoting sustainable farming, because it is clear that the damage that beetles could cause is minimal and that biodiversity can be preserved.

Introduction. I missed a more detailed analysis of insect nutrition based on entomological articles, works by well-known entomologists, and family revisions. In particular, I have observed that in some entomological sources, Silpha tristis is presented as carrion beetles, so if this is true, then the interpretation of finding the largest group in these studies should change substantially. A more detailed review of other insect nutrition research articles in this article would be needed (whether these are random observations or detailed and in-depth nutrition research has been conducted). It is not clear what size the snails could be that the beetles could eat.

You will not change the design of the study, but it would still be more reliable, if possible, to compare data from plots of the same area than results from sampled surfaces, which are now twice as different, especially since the total number of beetles is presented later. numbers.

The description of the methods is null and void: 1) it is not clear at what time the tests were performed on different farms - they may differ, then the results are unreliable (beetle abundance will vary depending on their biological cycle) - exact dates are needed; 2) it is not clear how the uniformity of the samples was ensured in different farms - if the survey area is larger, if the number of samples is higher, the frequency of beetle detection may be higher - the exact number of samples in different farms is needed; 3) the search methodology itself remains unclear - it may vary from farm to farm - simple differences in duration may already skew the results; 4) it is not clear who, how, according to what literature characterized beetles.

Although we see pictures of snail farms, we see nets - it is not clear how they are sealed or whether the nets are dense enough to prevent insects from entering.

Although it was mentioned in the methodological part that the completeness and consistency of the collected data were evaluated, this is not reflected in the presentation of the results. It is also unclear where the Chi-square or the Fisher, Kruskal – Wallis tests were actually used - it is not clear in the results whether there are reliable differences between farms, species, seasons.

Given the weakness of the methodological and statistical analysis, I would suggest that the article should simply be redirected to a general analysis of bettles diversity, their ecological groups, without orientation to damage of beetles or protection of snails.

Author Response

  • I think this article would be interesting for readers, not so much for snail farmers, but more for those interested in tackling mollusc damage in agrolandscapes in search of effective biological control measures. I suggest a complete change in the idea of the article, from avoiding damage to snails on farms to promoting sustainable farming, because it is clear that the damage that beetles could cause is minimal and that biodiversity can be preserved.
    • R: dear review, we agree with your interest and maybe this would a future scope for our work. Otherwise, this research (HelixREC project (https://www.izs-sardegna.it/helixrec/) is focused from the part of the snail’ farmers. Currently, it is not possibile to change the idea of the project considering the amount of fundings received, but certanly we’ll take care about this idea for the future.

  • I missed a more detailed analysis of insect nutrition based on entomological articles, works by well-known entomologists, and family revisions. In particular, I have observed that in some entomological sources, Silpha tristis is presented as carrion beetles, so if this is true, then the interpretation of finding the largest group in these studies should change substantially. A more detailed review of other insect nutrition research articles in this article would be needed (whether these are random observations or detailed and in-depth nutrition research has been conducted). It is not clear what size the snails could be that the beetles could eat.
    • R: dear review, the Introduction has been completely modified following your suggestions, including more biological information, in deep information about Silpha tristis has been added in lines 284-287, as well as those related to the dimension of the snails (lines 320-323)

  • You will not change the design of the study, but it would still be more reliable, if possible, to compare data from plots of the same area than results from sampled surfaces, which are now twice as different, especially since the total number of beetles is presented later numbers.
    • R: the statistical analyses were updated comparing all those features collected, following your suggestions and those from the other reviewers. Thank you very much.
  • The description of the methods is null and void: 1) it is not clear at what time the tests were performed on different farms - they may differ, then the results are unreliable (beetle abundance will vary depending on their biological cycle) - exact dates are needed; 2) it is not clear how the uniformity of the samples was ensured in different farms - if the survey area is larger, if the number of samples is higher, the frequency of beetle detection may be higher - the exact number of samples in different farms is needed; 3) the search methodology itself remains unclear - it may vary from farm to farm - simple differences in duration may already skew the results; 4) it is not clear who, how, according to what literature characterized beetles.
    • R: Following all your suggestions, the material and methods have been completely revised, as presented in lines 113-217, including: specific data of the samples 1) it is not clear at what time the tests were performed on different farms - they may differ, then the results are unreliable (beetle abundance will vary depending on their biological cycle) - exact dates are needed; 2) it is not clear how the uniformity of the samples was ensured in different farms - if the survey area is larger, if the number of samples is higher, the frequency of beetle detection may be higher - the exact number of samples in different farms is needed; 3) the search methodology itself remains unclear - it may vary from farm to farm - simple differences in duration may already skew the results; 4) it is not clear who, how, according to what literature characterized beetles

  • Although we see pictures of snail farms, we see nets - it is not clear how they are sealed or whether

the nets are dense enough to prevent insects from entering.

  • R: description of the farms has been updated in lines 129-152

  • Although it was mentioned in the methodological part that the completeness and consistency of the collected data were evaluated, this is not reflected in the presentation of the results. It is also unclear where the Chi-square or the Fisher, Kruskal – Wallis tests were actually used - it is not clear in the results whether there are reliable differences between farms, species, seasons.
    • Dear review, considering the very low number of farms included in this study (7), inference analysis is not strongly suggested. Otherwise, following your comments, the statistical analysis paragraph was added in material and methods and in results. Furthermore, statistical analysis was carried out to first investigate the possibile factors associated with the increasing number of insect detected.

  • Given the weakness of the methodological and statistical analysis, I would suggest that the article should simply be redirected to a general analysis of bettles diversity, their ecological groups, without orientation to damage of beetles or protection of snails.
    • R: please, see previous answer. The statistical analysis paragraph was added in material and methods and in results. Furthermore, statistical analysis was carried out to first investigate the possible factors associated with the increasing number of insect detected.

Round 2

Reviewer 1 Report

Dear authors. I am grateful for your answers to my comments. You have answered many of the comments and I can accept these answers. However, the main question was asked about sampling methods. In order to obtain high-quality results of their research, it was necessary to think over different methods of sampling beforehand. If some such methods are impossible due to various reasons, then others may be applicable. For example, collecting insects for light or for different baits, different types of traps, etc. At the same time, you do not have these methods of collecting samples and not enough material has been obtained for research. Therefore, I believe that the research should be continued and only after collecting a large amount of material can the manuscript be prepared.

Author Response

Dear review,
thank you for your suggestions, we partially agree with you, but as you can imagine, the aims of this work are not so simple given that this is the first application in such arguments, at least in Italy where the snails breeding is becoming even more frequent.
We strongly believe in this work, thus we tried to follow all your suggestions, even those of including more data about the current season. 
Furthermore, we strongly implement the discussion including reference on the statistical analyses.
We took trucks in word file only of these new revisions, accepting all those previous to avoid confusion.
Many thanks for your support.
Beste regards,
Federica Loi

Reviewer 2 Report

The manuscript is well done however it will improve a lot if the authors could wait to add the data of the present season if they can.

Author Response

Dear review,
first, I would like to thank you from myself and authors to your trust on this work.
As you can imagine, the aims of this work are not so simple given that this is the first application in such arguments, at least in Italy where the snails breeding is becoming even more frequent.
We strongly believe in this work, thus we tried to follow all your suggestions, even those of including more data about the current season. 
Furthermore, we strongly implement the discussion including reference on the statistical analyses.
We took trucks in word file only of these new revisions, accepting all those previous to avoid confusion.
Many thanks for your support.
Beste regards,
Federica Loi

Round 3

Reviewer 1 Report

Dear authors. Thanks for the answers. You have substantially corrected the manuscript. Continuation of experiments is necessary to obtain more adequate results.